# Effects of high intensity interval training versus moderate intensity continuous training on exercise capacity and quality of life in patients with heart failure: A systematic review and meta-analysis

**Shengyuan Gu[1], Xinchao Du[1]\*, Dongwei Wang[2], Yaohua Yu[3], Shifang Guo[3]**

**1** Department of Cardiology, Zhengzhou Central Hospital Affiliated to Zhengzhou University, Zhengzhou, Henan, China, **2** Department of Cardiac Rehabilitation, Zhengzhou Central Hospital Affiliated to Zhengzhou University, Zhengzhou, Henan, China, **3** Department of Respiratory Medicine and Pulmonary Rehabilitation, Zhengzhou Central Hospital Affiliated to Zhengzhou University, Zhengzhou, Henan, China

\* duthingchao@126.com

**Data Availability Statement:** All relevant data are within the manuscript and its Supporting Information files.

## Abstract

### Introduction and aims

High intensity interval training (HIIT) is considered as an alternative exercise modality to moderate intensity continuous training (MICT) for heart failure (HF) patients. Yet a growing number of trials demonstrated inconsistent findings about the effectiveness of HIIT versus MICT until SMARTEX study and OptimEx-Clin study have made a consistent negative conclusion that HIIT was not superior to MICT. The aim of this study was to conduct a meta-analysis involving a subgroup analysis of total exercise time (TET) and disease categories of HF to investigate if TET could affect the superiority of HIIT when compared with MICT.

### Methods and results

An electronic literature search of Pubmed, Embase, Cochrane Central Register of Controlled Trials and ClinicalTrials.gov was performed for this review. 16 studies of 661 patients were finally pooled into quantitative synthesis. The weighted mean difference (WMD) and standard mean difference (SMD) with 95% confidence interval (CI) were calculated for quantitative synthesis of outcomes. HIIT was superior to MICT in improving peak oxygen consumption (Peak VO$_2$) (WMD: 1.68 ml · kg$^{-1}$ · min$^{-1}$ 95% CI: 0.81 to 2.55 n = 661). The subgroup analysis of TET showed that HIIT was superior to MICT in improving Peak VO$_2$ in "short time" subgroup (WMD: 1.61 ml · kg$^{-1}$ · min$^{-1}$ 95% CI: 0.45 to 2.77 n = 166) and in "medium time" subgroup (WMD: 1.74 ml · kg$^{-1}$ · min$^{-1}$ 95% CI: 0.53 to 2.95 n = 420), and that there was no significant difference between HIIT and MICT in improving Peak VO$_2$ in "long time" subgroup (WMD: 0.62 ml · kg$^{-1}$ · min$^{-1}$ 95% CI: -1.34 to 2.58 n = 75).

**Funding:** The authors received no specific funding for this work.

**Competing interests:** The authors have declared that no competing interests exist.

## Conclusions

The superiority of HIIT to MICT in improving Peak $VO_2$ arose in a short to medium length of TET whereas it was effaced by an increment of TET. This "paradox" of TET on HIIT versus MICT might be due to the increasing poor adherence to target exercise intensity over time.

## Trial registration

**PROSPERO registration number:** CRD42022375076.

## 1. Introduction

Heart failure (HF) is a common, disabling, and deadly disease with a prevalence about 40 million people worldwide and it leads to a poor quality of life (QoL) under the numerous symptoms including shortness of breath, weakness and peripheral edema due to the stiffness or weakness of the heart muscle. In general, the global incidence of HF ranges from 100 to 900 cases per 100,000 person-years and the estimates of HF prevalence of developed countries range from 1–2% of the adult population. HF contributes to the increasing mortality and hospitalization, which becomes a heavy burden to healthcare and society. Hospitalization of HF represents 1–2% of all hospital admissions. Of note, high-income countries, which constitute only 18% of the global population, have contributed to 86% of the worldwide expenditure for HF. A study published in 2014 estimated that the global expenditure of 2012 for HF was 108 billion dollars, of which 60% was spent directly on medical costs [1–3].

Exercise training (ET) is beneficial to HF patients besides the optimal medical therapy [4–6], and moderate intensity continuous training (MICT) is regarded as the standard exercise modality for patients with HF in ET programs [7]. Recently, high intensity interval training (HIIT) is considered as an alternative exercise modality to MICT for HF patients [8–10]. However, it remains unknown which one of the HIIT and MICT is the optimal training modality for HF Patients, and thus a growing number of randomized trials were designed to compare the effectiveness of HIIT versus MICT on HF patients [11]. Yet the findings keep inconsistent until the multicenter large sample studies, SMARTEX Heart Failure study [7] and OptimEx-Clin study [9], have been published. SMARTEX Heart Failure study on patients of heart failure with reduced ejection fraction (HFrEF) reported that HIIT was not superior to MICT in changing aerobic capacity, meanwhile OptimEx-Clin study on patients of heart failure with preserved ejection fraction (HFpEF) concluded that there was no statistically significant difference in change of peak oxygen consumption (Peak $VO_2$) at 3 months between HIIT and MICT. As if the results of the two studies would calm down the different sounds about the effectiveness of HIIT versus MICT. Of note, both of the two studies have referred to the attenuation in adherence to target exercise intensity or exercise protocols and expressed the concern that poor adherence might affect the results. Thus, in order to make it clear if an "overlong" exercise time, through the attenuation in adherence which might get more serious over time, has indirectly affected these "negative" results, a meta-analysis involving a subgroup analysis of exercise time becomes necessary. As we all know, training duration should not completely represent the real exercise time, for example, a shorter training duration might have a longer total exercise time (TET). A regular exercise prescription defines a duration only by days, weeks or months, but seldom by minutes. While TET, which is defined by the volume of time of total exercise session and calculated by minutes, provides a more comprehensive measure than

duration, when estimating the training effectiveness which was produced through exercise intensity and total exercise time rather than through exercise intensity and exercise duration. Thus, from our perspective, it seems better to conduct a subgroup meta-analysis by gathering studies with similar TET together than with similar training duration, when estimating the effectiveness of HIIT versus MICT. Moreover, Ismail et al. [12] have reported that TET may be a confounder in their meta-analysis in regard to the cardiovascular responses to different exercise training in HF patients. However, the effect of TET on HIIT versus MICT has never been investigated in meta-analysis involving HF patients.

As is well known, HF is divided into two groups by left ventricular ejection fraction (LVEF), including HFrEF and HFpEF. Although these two disease entities have several same symptoms, their etiological and epidemiological profiles still differ with each other [13, 14]. Thus, it is necessary to conduct a subgroup analysis when a systematic review of HF is performed. Cardiopulmonary exercise testing (CPET) is recommended by guidelines to assess overall cardiovascular health and performance [7]. Besides, Peak $VO_2$ and the minute ventilation to carbon dioxide production slope (VE/$VCO_2$ slope) are defined as the key variables of CPET, and Peak $VO_2$ is the best variable to assess physical fitness [15, 16]. As a patient-centered outcome, QoL is also an important endpoint in assessment of effects of ET [17].

Taking all those discussed above into account, this study aimed at conducting a meta-analysis assessing the effects of HIIT versus MICT on Peak $VO_2$, VE/$VCO_2$ slope and QoL in HF patients and involving a subgroup analysis of TET and disease categories of HF, in order to investigate if TET could affect the superiority of HIIT when compared with MICT.

### 1.1. Review question

Has an "overlong" exercise time, through the attenuation in adherence which might get more serious over time, indirectly contributed to no significant difference between HIIT and MICT in change of Peak $VO_2$?

### 1.2. Eligibility criteria

This systematic review included studies on the effects of HIIT compared to MICT in HF patients. To be eligible, the trial had to include patients with HF randomized to either to HIIT or MICT.

The inclusion criteria were as follows: a) including adult patients (aged ≥18 years); b) a randomized controlled trial. The exclusion criteria were as follows: a) articles in non-English; b) not being high versus moderate intensity at the interval training group and the continuous training group; c) lack of primary outcome; d) data of primary outcome not available or complete; e) data from the same trial. The primary outcome of interest was Peak $VO_2$ and the other outcomes were VE/$VCO_2$ slope and QoL.

## 2. Methods

This meta-analysis was registered on the PROSPERO website and performed under the recommendations of Preferred Reporting Items for Systematic Reviews and Meta-Analyses (PRISMA) [18].

### 2.1. Search strategy

A systematic literature search of Pubmed, Embase, Cochrane Central Register of Controlled Trials and ClinicalTrials.gov (from the earliest date available to November 2022) was

performed independently by two reviewers. Three groups of keywords including the medical subject headings and their related terms were combined for searching (as shown in S1 File).

## 2.2. Screening and selection

After duplicates removed, two independent reviewers screened all titles and abstracts to identify potentially eligible studies that compared continuous training with interval training or HIIT in HF patients. And then the articles were retrieved for full text and independently assessed by the two reviewers if they met the criteria for eligibility. References of retrieved articles were checked manually for other potentially eligible studies. An advisory group, which consisted of the other three authors except the two independent reviewers, was set to solve any disagreement in the process of screening and selecting studies.

## 2.3. Critical appraisal

Methodological quality of included studies was independently assessed by two authors using the Cochrane risk of bias (Cochrane ROB) tool through the judgement of bias which includes selection bias of random sequence generation and allocation concealment, performance bias of blinding of participants and personnel, detection bias of blinding of outcome assessment, attrition bias of incomplete outcome data, reporting bias of selective reporting and other bias. Any disagreement was resolved by discussion with the advisory group.

## 2.4. Data extraction

The data was extracted independently by two reviewers using standard data extraction forms of Cochrane guidelines (Higgins and Green 2009) for the first author, publication year, country, characteristics of population, diagnosis of disease, LVEF, characteristics of intervention, outcomes at baseline and at the end of the follow-up of the supervised cardiac rehabilitation. If exercise time of per session was not reported directly in the article, it was calculated by the details of training protocol (including warm-up and cool-down). For continuous variables, results were expressed as the mean difference of the variable between randomized groups. Methods established by Luo et al. [19] and Wan et al. [20] were used to obtain the mean and standard deviation of the continuous data of Peak $VO_2$, VE/$VCO_2$ slope and QoL.

## 2.5. Data synthesis

We analyzed pooled studies in a random-effects model using the statistical method of Inverse Variance with Review Manager, and calculated the weighted mean difference (WMD) or standard mean difference (SMD) with 95% confidence interval (CI). Heterogeneity was quantified with Review Manager using the $I^2$ statistic; $I^2$ values of 25%, 50%, and 75% represented low, moderate, and high heterogeneity, respectively. Sensitivity analyses were performed using the statistical method of meta-based Influence Analysis with STATA to assess the general effects after specific studies were omitted. A descriptive statistics analysis of TET, which included normality tests, percentiles distribution and frequency histogram, was conducted with SPSS for dividing groups of studies into "short time", "medium time" and "long time". Subgroup analyses of the TET and disease categories (HFrEF and HFpEF) were performed with Review Manager to assess part of potential source of heterogeneity. A funnel plot and contour-enhanced funnel plot were made with STATA to identify publication bias. A two-tailed $P$ value lower than 0.05 was considered statistically significant. Review Manager (Version 5.4), SPSS (Version 21.0) and STATA (Version 16.0) were the software used in the analyses.

## 3. Results

### 3.1. Study inclusion

459 articles were identified from the initial search of four databases. Out of the 459 studies, 199 duplicates were identified and another 232 studies were excluded after screening the title and abstract. Among the 232 studies, 46 were reviews; 21 were animal experiments; 1 was case report; 40 were conference abstracts without full text available; 57 were protocols; 17 were sub-studies; 1 was guideline; 1 was hypothesis; 48 were studies which did not meet the eligibility criteria obviously after screening the title and abstract. Then 28 articles were retrieved for eligibility, and 12 articles were excluded and 16 articles were included after the assessment of full text. Of the 12 articles, 1 was article in German; 2 lacked primary outcome; 1 lacked data of primary outcome; 3 had an equal exercise intensity in two groups; 5 had the same date derived from the same trial. The PRISMA flowchart shows the process of literature search and filtering (Fig 1).

### 3.2. Characteristics of included studies and methodological quality

Of the 16 trials [8, 10, 21–34], 661 patients were involved in this meta-analysis with a range from 53.2 [33] to 76.5 [34] years old in mean age. Three studies [10, 21, 25] involved patients

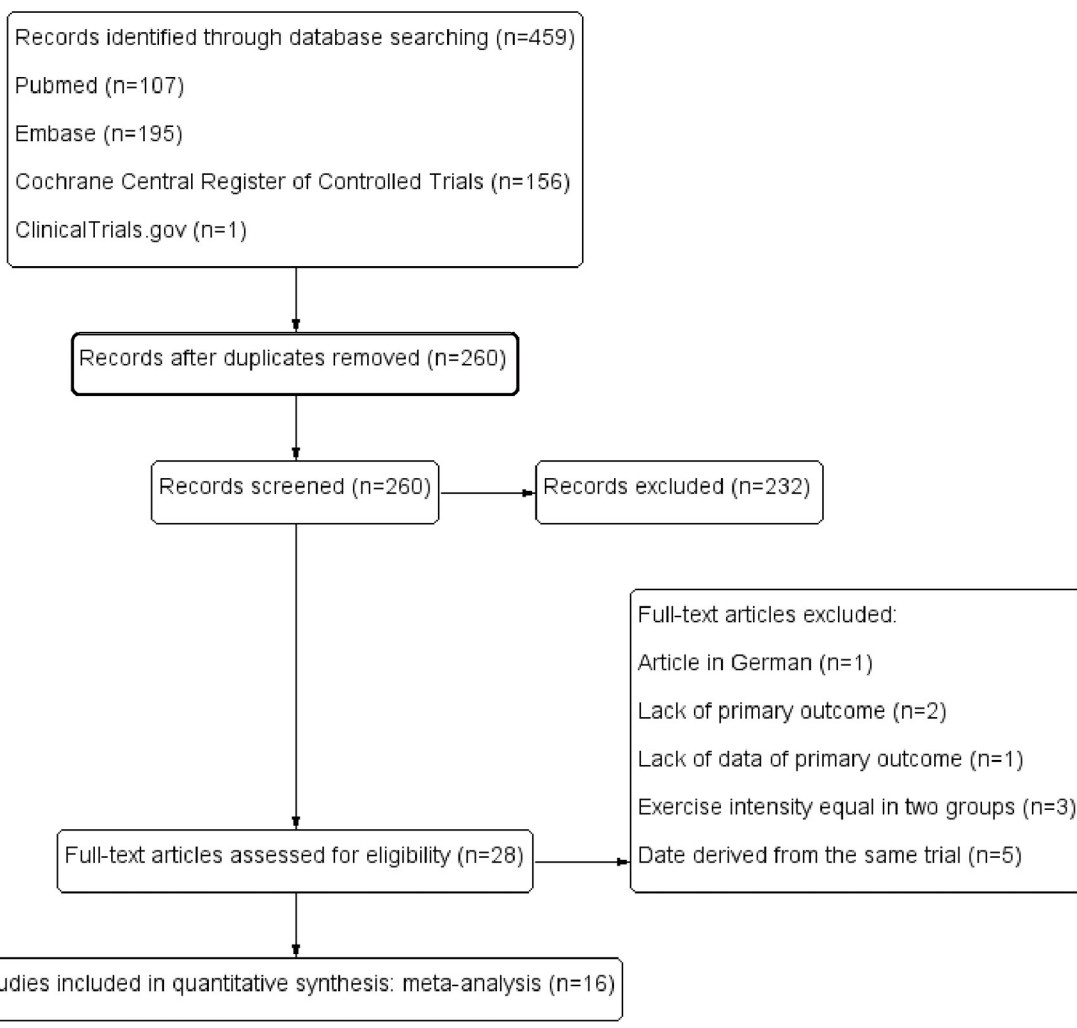

**Fig 1. Preferred Reporting Items for Systematic Reviews and Meta-Analyses (PRISMA) flow diagram.**

**Table 1. General characteristics of included studies.**

| Study | Year | Country | N, gender (male/female) | Mean age (year) | Diagnosis (NYHA) | LVEF (%) | Peak VO$_2$ at baseline (ml · kg$^{-1}$ · min$^{-1}$) |
|---|---|---|---|---|---|---|---|
| 1 Angadi et al. | 2015 | America | HIIT: 9 M/F: 8/1 | 69 ± 6.1 | NR | 63.7 ± 6.4 | 19.2 ± 5.2 |
| | | | MICT: 6 M/F: 4/2 | 71.5 ± 11.7 | NR | 66.0 ± 4.7 | 16.9 ± 3.0 |
| 2 Benda et al. | 2015 | Netherlands | HIIT: 10 M/F: 9/1 | 63 ± 8 | II/III: 8/2 | 37 ± 6 | 19.1 ± 4.1 |
| | | | MICT: 10 M/F: 10/0 | 64 ± 8 | II/III: 8/2 | 38 ± 6 | 21.0 ± 3.4 |
| 3 Besnier et al. | 2019 | France | HIIT: 16 M/F: 11/5 | 59 ± 13 | I/II/III: 8/7/1 | 36 ± 8 | 17.2 ± 4.5 |
| | | | MICT: 15 M/F: 11/4 | 59.5 ± 12 | I/II/III: 6/7/2 | 36 ± 7 | 15.0 ± 4.6 |
| 4 Dimopoulos et al. | 2006 | Greece | HIIT: 10 M/F: 9/1 | 59.2 ± 12.2 | I/II/III: 3/6/1 | 34.5 ± 10.5 | 15.4 ± 4.7 |
| | | | MICT: 14 M/F: 14/0 | 61.5 ± 7.1 | I/II/III: 4/9/1 | 30.7 ± 10.3 | 15.5 ± 3.7 |
| 5 Donelli et al. | 2020 | Brazil | HIIT: 10 M/F: 3/7 | 60 ± 10 | II/III: 8/2 | 65 ± 5 | 16.1 ± 3.3 |
| | | | MICT: 9 M/F: 4/5 | 60 ± 9 | II/III: 8/1 | 65 ± 5 | 17.6 ± 3.5 |
| 6 Ellingsen et al. | 2017 | Norway | HIIT: 77 M/F: 63/14 | 65 ± 22.4 | II/III: 55/22 | 29 ± 11.2 | 16.8 ± 4.5 |
| | | | MICT: 65 M/F: 53/12 | 60 ± 14.4 | II/III: 41/24 | 29 ± 12.3 | 16.2 ± 7.0 |
| 7 Freyssin et al. | 2012 | France | HIIT: 12 M/F: 6/6 | 54 ± 9 | NR | 27.8 ± 4.7 | 10.7 ± 2.9 |
| | | | MICT: 14 M/F: 7/7 | 55 ± 12 | NR | 30.7 ± 7.8 | 10.6 ± 4.1 |
| 8 Fu et al. | 2013 | Taiwan | HIIT: 15 M/F: 10/5 | 67.5 ± 1.8 | NR | 38.3 ± 3.5 | 16.0 ± 1.0 |
| | | | MICT: 15 M/F: 9/6 | 66.3 ± 2.1 | NR | 38.6 ± 4.8 | 15.9 ± 0.7 |
| 9 Iellamo et al. | 2013 | Italy | HIIT: 8 M/F: NR | 62.2 ± 8 | NR | 31.5 ± 6.9 | 18.8 ± 4.6 |
| | | | MICT: 8 M/F: NR | 62.6 ± 9 | NR | 32.8 ± 6.5 | 18.4 ± 4.3 |
| 10 Iellamo et al. | 2014 | Italy | HIIT: 18 M/F: 16/2 | 67.2 ± 6 | I/II: 6/12 | 34.1 ± 6 | 15.3 ± 4 |
| | | | MICT: 18 M/F: 15/3 | 68.4 ± 8 | I/II: 8/10 | 35.6 ± 7 | 16.2 ± 5 |
| Study | Year | Country | N, gender (male/female) | Mean age (year) | Diagnosis (NYHA) | LVEF (%) | Peak VO$_2$ at baseline (ml · kg$^{-1}$ · min$^{-1}$) |
| 11 Koufaki et al. | 2014 | UK | HIIT: 16 M/F: 14/2 | 59.8 ± 7.4 | NR | 41.7 ± 10.3 | 14.6 ± 4.8 |
| | | | MICT: 17 M/F: 13/4 | 59.7 ± 10.8 | NR | 35.2 ± 6.4 | 15.5 ± 15.9 |
| 12 Mueller et al. | 2021 | Germany | HIIT: 58 M/F: 17/41 | 70 ± 7 | II/III: 44/14 | NR | 18.9 ± 5.4 |
| | | | MICT: 58 M/F: 23/35 | 70 ± 8 | II/III: 44/14 | NR | 18.2 ± 5.1 |
| 13 Papathanasiou et al. | 2020 | Bulgaria | HIIT: 60 M/F: 35/25 | 63.7 ± 6.7 | II/III: 48/12 | 35.9 ± 2.3 | 13.5 ± 3.8 |
| | | | MICT: 60 M/F: 35/25 | 63.8 ± 6.7 | II/III: 46/14 | 36.0 ± 2.0 | 12.5 ± 3.6 |
| 14 Roditis et al. | 2007 | Greece | HIIT: 11 M/F: 10/1 | 63 ± 2 | I/II/III: 3/7/1 | 30.7 ± 10.3 | 14.2 ± 3.1 |
| | | | MICT: 10 M/F: 9/1 | 61 ± 3 | I/II/III: 4/5/1 | 34.5 ± 10.5 | 15.3 ± 4.4 |
| 15 Ulbrich et al. | 2016 | Brazil | HIIT: 12 M/F: NR | 53.2 ± 7.0 | II/III: 11/1 | 35.4 ± 6.7 | 21.4 ± 4.1 |
| | | | MICT: 10 M/F: NR | 54.0 ± 9.9 | II: 10 | 32.8 ± 7.7 | 18.4 ± 4.3 |
| 16 Wisløff et al. | 2007 | Norway | HIIT: 9 M/F: 7/2 | 76.5 ± 9 | NR | 28.0 ± 7.3 | 13.0 ± 1.6 |
| | | | MICT: 9 M/F: 7/2 | 74.4 ± 12 | NR | 32.8 ± 4.8 | 13.0 ± 1.1 |

NYHA: New York Heart Association functional class of heart failure ranging from I to IV; Peak VO$_2$: peak oxygen consumption; NR: not reported.

of HFpEF while thirteen [8, 22–24, 26–34] recruited patients of HFrEF. The general characteristics of included studies are shown in Table 1. The characteristics of HIIT and MICT have been summarized in Table 2. Five studies [21–23, 27, 28] did not report directly the exercise time of per session. The results of the assessment of the risk of bias are presented in Fig 2.

### 3.3. Empirical results

**3.3.1. Total exercise time.** The normality tests of Shapiro-Wilk for TET expressed a normal distribution in both HIIT group and MICT group. The range of TET was divided into three equal portion, and the short part was defined as the range of "short-time" group, the medium part as the range of "medium-time" group and the long part as the range of "long-time" group (S2 File).

**Table 2. Intervention characteristics of HIIT versus MICT in included studies.**

| Study | Type | Intensity | Exercise time per session (min) | Duration (week) | Total sessions | Total exercise time (min) |
|---|---|---|---|---|---|---|
| 1 Angadi et al. | Running | HIIT: 1st week: 4×2 min 80–85% PHR; 4×2 min 50% PHR 2nd–4th week: 4×4 min 85–90% PHR; 3×3 min 50% PHR | 1st week: 31* 2nd–4th week: 40* | 4 | 12 | 453 |
| | | MICT: 1st week: 60% PHR 2nd–4th week: 70% PHR | 1st week: 30* 2nd–4th week: 45* | | | 495 |
| 2 Benda et al. | Cycling | HIIT: 10×1 min 90% Watt; 10×2.5 min 30% Watt | 50* | 12 | 24 | 1200 |
| | | MICT: 60–75% Watt | 45* | | | 1080 |
| 3 Besnier et al. | Cycling | HIIT: 2×8 min interval training (8×0.5 min 100% PPO alternating with 8×0.5 min 50% PPO); 1×4 min passive recovery | 30* | 3.5 | 18 | 540 |
| | | MICT: 60% PPO | 40* | | | 720 |
| 4 Dimopoulos et al. | Cycling | HIIT: alternating 0.5 min 100% pWR with 0.5 min rest | 40 | 12 | 36 | 1440 |
| | | MICT: 50% pWR | 40 | | | 1440 |
| 5 Donelli et al. | Running | HIIT: 4×4 min 80–90% Peak VO$_2$; Three intervals at 50–60% Peak VO$_2$ | 38 | 12 | 36 | 1368 |
| | | MICT: 50–60% Peak VO$_2$ | 47 | | | 1692 |
| 6 Ellingsen et al. | Running or Cycling | HIIT: 4×4 min 90%–95% PHR; 3×3 min 50–70% PHR | 38 | 12 | 36 | 1368 |
| | | MICT: 60%–70% PHR | 47 | | | 1692 |
| 7 Freyssin et al. | Running or Cycling | HIIT: 1st–4th week: 3×12 repetitions of interval training (0.5 min 50% maximal power alternating with 1 min rest); 2×5 min rest 5th–8th week: 3×12 repetitions of interval training (0.5 min 100% maximal power alternating with 1 min rest); 2×5 min rest | 28 | 8 | 48 | 1344 |
| | | MICT: HR at the VT1 | 60 | | | 2880 |
| Study | Type | Intensity | Exercise time per session (min) | Duration (week) | Total sessions | Total exercise time (min) |
| 8 Fu et al. | Cycling | HIIT: 5×3 min 80% Peak VO$_2$; 4×3 min 40% Peak VO$_2$ | 33* | 12 | 36 | 1188 |
| | | MICT: 60% Peak VO$_2$ | 36* | | | 1296 |
| 9 Iellamo et al., 2013 | Uphill walking | HIIT: 4×4 min 75–80% HRR; 4×3 min 50% HRR | Week 1–3, 3–6, 6–9, 9–12 23, 30, 37, 37 | 12 | 42 | 1407 |
| | | MICT: 45–60% HRR | 30, 35, 40, 45 | | | 1650 |
| 10 Iellamo et al., 2014 | Uphill walking | HIIT: 4×4 min 75–80% HRR; 4×3 min 45–50% HRR | 48* | 12 | 36 | 1728 |
| | | MICT: 45–60% HRR | 50–65* | | | 1800–2340 |
| 11 Koufaki et al. | Cycling | HIIT: 2×15 min cycling (1 min 20–30% PPO alternating with 0.5 min 100% PPO) | 30–34 | 24 | 72 | 2160–2448 |
| | | MICT: 40–60% Peak VO$_2$ | 27–40 | | | 1944–2880 |
| 12 Mueller et al. | Cycling | HIIT: 4×4 min 80%–90% HRR; 3×4 min active recovery | 38 | 12 | 36 | 1368 |
| | | MICT: 35%–50% HRR | 40 | | 60 | 2400 |
| 13 Papathanasiou et al | Cycling | HIIT: three high intensity intervals at 90% HR$_{max}$; two moderate-intensity intervals at 70% HR$_{max}$ | 40 | 12 | 24 | 960 |
| | | MICT: 70% HR$_{max}$ | 40 | | | 960 |
| 14 Roditis et al. | Cycling | HIIT: 0.5min 100% pWR alternating with 0.5 min rest | 40 | 12 | 36 | 1440 |
| | | MICT: 50% pWR | 40 | | | 1440 |
| 15 Ulbrich et al. | Uphill walking or running | HIIT: 3-min intervals at 95% PHR interspersed by active recovery of 70% PHR | 60 | 12 | 36 | 2160 |
| | | MICT: 75% PHR | 60 | | | 2160 |

(*Continued*)

**Table 2.** (Continued)

| Study | Type | Intensity | Exercise time per session (min) | Duration (week) | Total sessions | Total exercise time (min) |
|---|---|---|---|---|---|---|
| 16 Wisløff et al. | Uphill walking | HIIT: 4×4 min 90–95% PHR; 3×3 min 50%–70% PHR | 38 | 12 | 36 | 1368 |
| | | MICT: 70–75% PHR | 47 | | | 1692 |

*: exercise time calculated by the details of training protocol (including warm-up and cool-down); PHR: peak heart rate; Watt: maximal workload; PPO: peak power output; pWR: peak work rate; Peak VO$_2$: peak oxygen consumption; HR: heart rate; VT1: first ventilatory threshold; HRR: heart rate reserve; HR$_{max}$: maximum heart rate.

**3.3.2. Peak VO$_2$.** *3.3.2.1. Peak VO$_2$ in HF.* In HF studies [8, 10, 21–34], HIIT had a significant improvement in peak VO$_2$ of 1.68 ml · kg$^{-1}$ · min$^{-1}$ (95% CI: 0.81 to 2.55 n = 661) compared to MICT (Fig 3). Sensitive analyses (S1 Fig) showed that after omitting the included studies one by one, the lack of each of the three studies [10, 27, 34] led to an apparent change of the effectiveness of peak VO$_2$, which indicated that the three studies were the potential source of heterogeneity among HF studies. And likewise, the two studies [27, 34] were the potential source of heterogeneity among HFrEF studies. After exclusion of the three studies [10, 27, 34], the overall $I^2$ value reduced from 48% to 0%, which in turn proved that it was appropriate to identify the three studies as potential source of heterogeneity (Fig 4).

Subgroup analyses of TET of HIIT are shown in Fig 4A. After omitting three studies [10, 27, 34], the heterogeneity was significantly reduced in the "medium-time" subgroup (Fig 4B). As a result of omitting the three studies as potential source of heterogeneity, the change of peak VO$_2$ of "medium-time" group was reduced from 1.74 ml · kg$^{-1}$ · min$^{-1}$ (95% CI: 0.53 to 2.95 n = 420) to 1.05 ml · kg$^{-1}$ · min$^{-1}$ (95% CI: -0.15 to 2.25 n = 268). Subgroup analyses of TET of MICT showed the similar results (Fig 5).

*3.3.2.2. Peak VO$_2$ in HFrEF and HFpEF.* Subgroup analyses of disease categories showed HIIT increased peak VO$_2$ of 1.89 ml · kg$^{-1}$ · min$^{-1}$ (95% CI: 0.97 to 2.82 n = 520) comparing to MICT in HFrEF. And there was no significant difference between HIIT and MICT in improving peak VO$_2$ of 0.53 ml · kg$^{-1}$ · min$^{-1}$ (95% CI: -1.13 to 2.18 n = 141) in HFpEF (Fig 6).

Subgroup analyses of TET of HFrEF studies [8, 22–24, 26–34] were performed with the similar results to HF studies [8, 10, 21–34] (S2 and S3 Figs). HIIT was superior to MICT in improving Peak VO$_2$ in "short-time" group analysis of HIIT of HFrEF (WMD: 1.59 ml · kg$^{-1}$ · min$^{-1}$ 95% CI: 0.39 to 2.79 n = 151) and in "medium time" group analysis of HIIT of HFrEF (WMD: 2.17 ml · kg$^{-1}$ · min$^{-1}$ 95% CI: 0.94 to 3.39 n = 294), and there was no significant difference between HIIT and MICT in improving Peak VO$_2$ in "long-time" group analysis of HIIT of HFrEF (WMD: 0.62 ml · kg$^{-1}$ · min$^{-1}$ 95% CI: -1.34 to 2.58 n = 75) (S2 Fig).

HIIT was superior to MICT in improving Peak VO$_2$ in "short-time" group analysis of MICT of HFrEF (WMD: 2.37 ml · kg$^{-1}$ · min$^{-1}$ 95% CI: 0.99 to 3.76 n = 198), and there was no significant difference between HIIT and MICT in improving Peak VO$_2$ in "medium-time" group analysis of MICT of HFrEF (WMD: 1.18 ml · kg$^{-1}$ · min$^{-1}$ 95% CI: -0.70 to 3.06 n = 257) and in "long-time" group analysis of MICT of HFrEF (WMD: 1.92 ml · kg$^{-1}$ · min$^{-1}$ 95% CI: -0.11 to 3.94 n = 65) (S3 Fig).

**3.3.3. VE/VCO$_2$ slope and QoL.** Nine studies [10, 21, 22, 24, 25, 27–29, 32] reported VE/VCO$_2$ slope as outcome with 143 participants in HIIT and 142 in MICT. The meta-analysis showed no significant difference between HIIT and MICT in VE/VCO$_2$ slope of -0.76 (95% CI: -2.19 to 0.66 n = 285) (S4 Fig). There was no statistical difference between HIIT and MICT in QoL of -0.16 (95% CI: -0.48 to 0.17 n = 519) (S5 Fig).

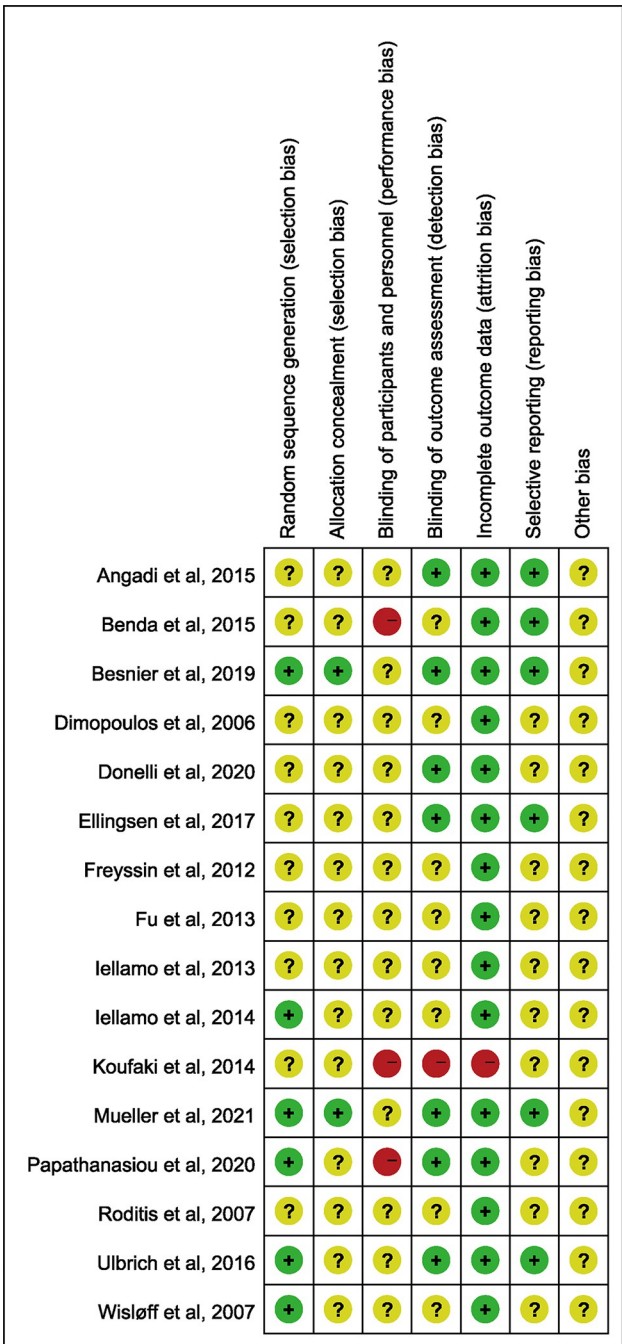

**Fig 2. Risk of bias summary.**

**3.3.4. Bias of publication.** The asymmetry of distribution of the studies in funnel plot showed that there might be a bias of publication. However, the further results of contour-enhanced funnel plot showed that 7 studies of the 8 imputed studies had a *P* value of lower than 0.05, which meant that there were other potential reasons, for example the different intervention characteristics of HIIT and MICT or the obvious difference between studies in the general characteristics including the ratio of men to women, mean age, and LVEF and Peak

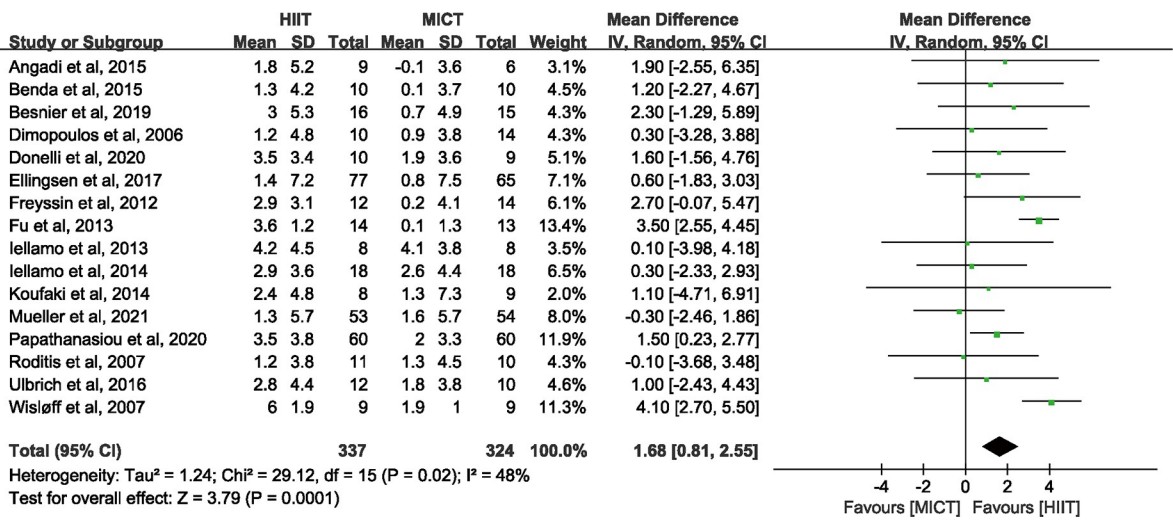

**Fig 3. Peak VO$_2$ in HF.**

VO$_2$ at baseline et al. (as shown in Tables 1 and 2), rather than the bias of publication contributing to the asymmetry of distribution (Fig 7).

## 4. Discussion

### 4.1. Discussion

Our systematic review demonstrated the three findings below. (1) The superiority of HIIT to MICT in increasing Peak VO$_2$ for HF patients appeared in a short to medium TET whereas it was effaced by an increment of TET. And this trend could also be applicable to HFrEF patients. (2) HIIT was not superior to MICT for HFpEF patients in improving Peak VO$_2$. (3) It did not support HIIT was more beneficial than MICT in VE/VCO$_2$ slope and QoL for HF patients.

In the past, MICT, as a preference in exercise prescription, was recommended for patients with HF [35]. Yet, after HIIT was introduced in cardiac rehabilitation by the Norwegian group of Wisloff et al. showing superior clinical improvements in HF patients [34], a strong clinical interest of HIIT versus MICT has emerged. Since then, a growing number of small individual studies have compared HIIT with MICT for HF patients on the effectiveness, but the results of them have remained inconsistent and inconclusive for a long time. Until the only two multicenter large sample studies, SMARTEX Heart Failure study [8] and OptimEx-Clin study [10], were published, it seems that the different sounds should have calmed down. SMARTEX Heart Failure study for HFrEF patients reported that there was no difference between HIIT and MICT in peak oxygen uptake, but both of them were superior to recommendation of regular exercise (RRE), and this superiority disappeared at the follow-up after 52 weeks. OptimEx-Clin study for HFpEF patients concluded that there was no statistically significant difference between HIIT and MICT in peak oxygen uptake at 3 months and neither group met the pre-specified minimal clinically important difference compared with the guideline control. Of note, in the sense of statistic, both HIIT and MICT actually had significant difference when compared with the guideline control at 3 months, and this significance finally disappeared at 12 months. In some sense, the two multicenter studies made a consistent conclusion. More importantly, both of the two studies have specifically expressed the worry that the attenuation in adherence to target exercise intensity or exercise protocols might affect their results. So it seems to be necessary to discuss if the flowing time, behind which was the attenuation in

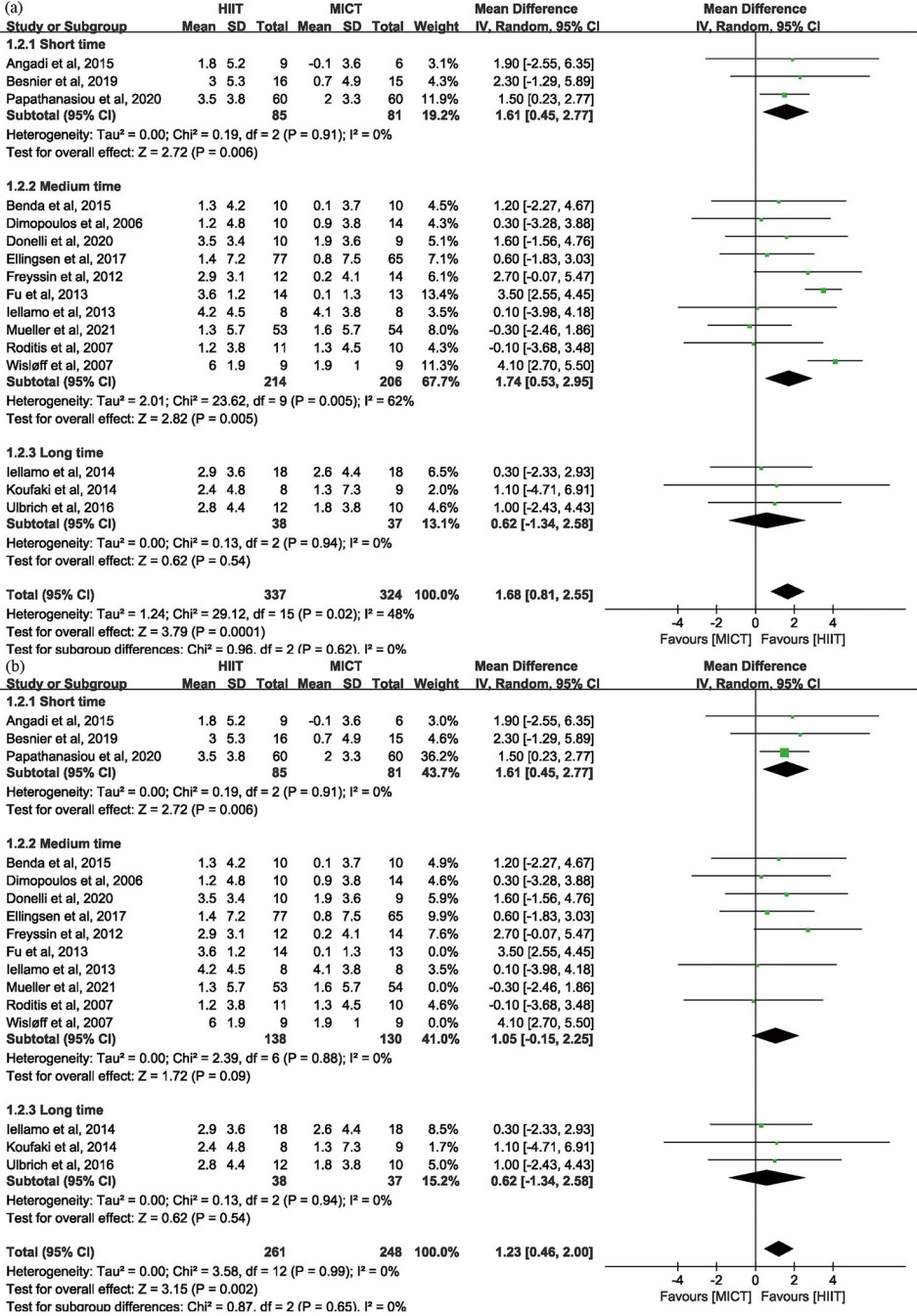

**Fig 4.** (a) Subgroup analysis of TET of HIIT for HF in Peak VO$_2$; (b) sensitive analysis.

adherence, had contributed to such "negative" results. Our results of this meta-analysis have given a "yes" answer to this question.

The novelty of this systematic review is that we conducted a subgroup analysis by gathering studies with similar TET together. Interestingly, the subgroup analysis showed that the TET was a confounder impacting the effectiveness of HIIT versus MICT in Peak VO$_2$; but it was paradoxical that an increment of TET effaced the superiority of HIIT rather than

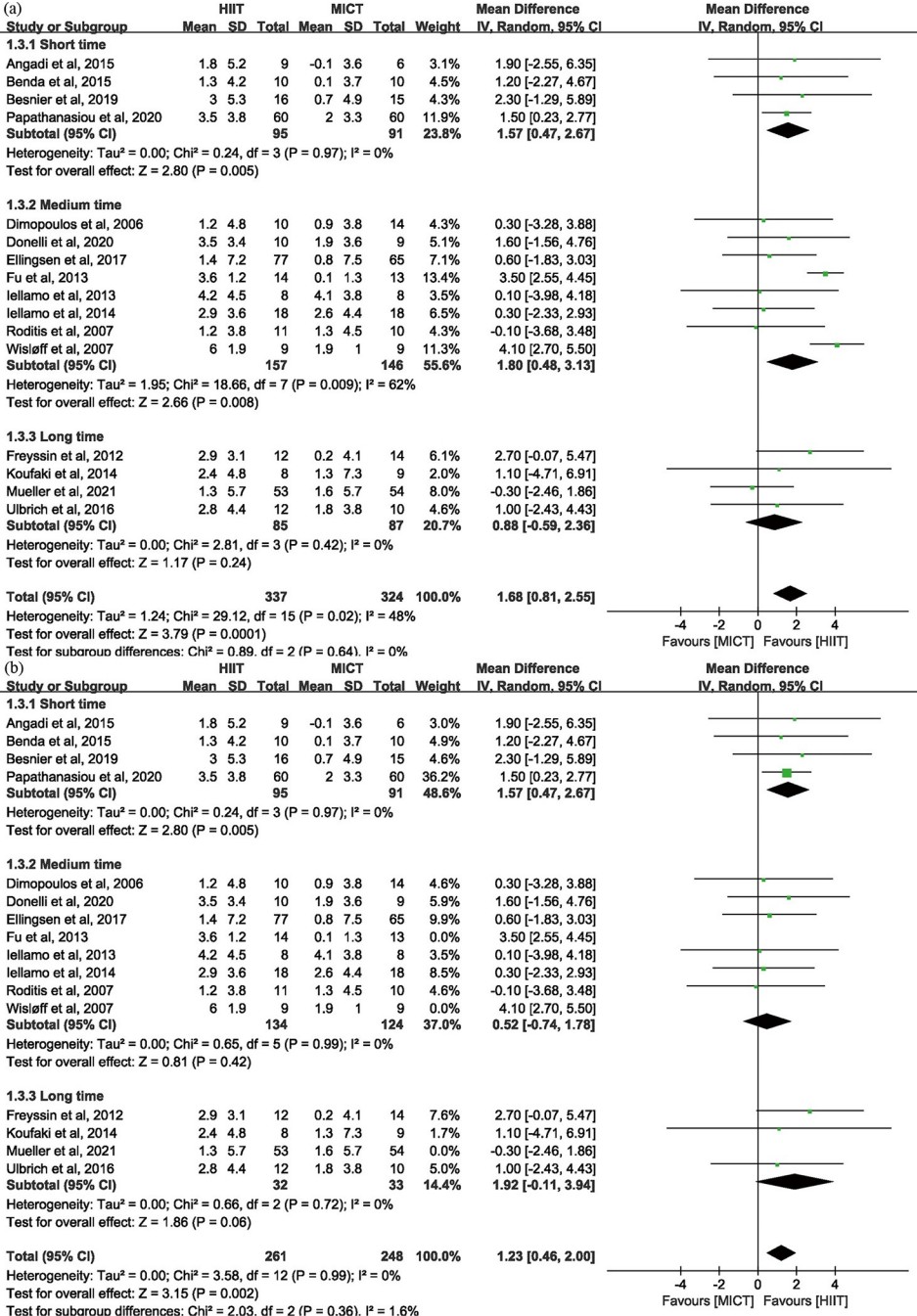

**Fig 5.** (a) Subgroup analysis of TET of MICT for HF in Peak VO$_2$; (b) sensitive analysis.

strengthening it. As is known, the "FITT" concept (frequency, intensity, time, and type) is regarded as the basic tenets in exercise prescription, and energy expenditure is the product of ET [6]. Exercise intensity is proportionally related to the training volume of each unit which in turn multiplied by TET provides the total energy expenditure of a training program, and so the superiority of HIIT to MICT should have been strengthened in the "long-time" subgroup. Thus a paradox of TET in a training program seems to appear. Likewise, the SMARTEX study

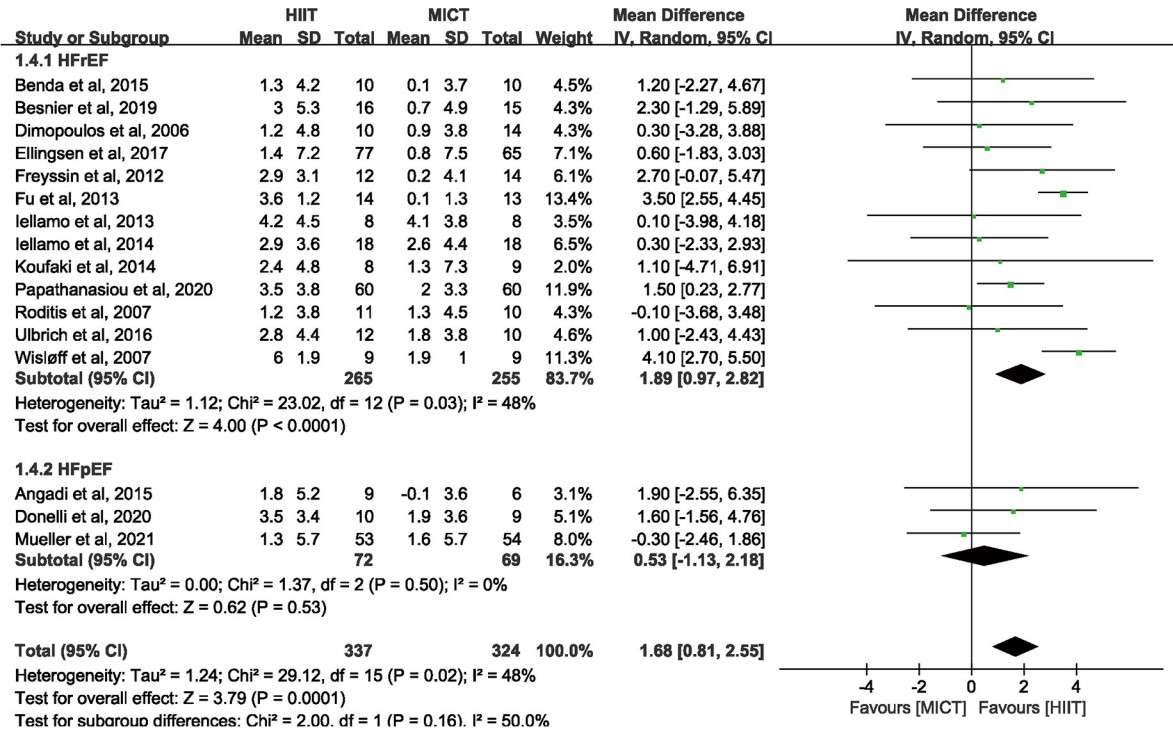

**Fig 6.** Subgroup analysis of disease categories of HFrEF and HFpEF in Peak VO$_2$.

[8] has drawn a notable conclusion that at the follow-up of week 12, HIIT and MICT were both superior to RRE, but the superiority disappeared at the follow-up of week 52. The training records of SMARTEX study revealed that 51% of patients in HIIT exercised below the target intensity during the supervised cardiac rehabilitation of 12 weeks versus 80% in MICT above the target intensity. It is reasonable to deem that the poor adherence in HIIT to the target intensity might become increasing over time and then the increasing poor adherence led to a similar energy expenditure between HIIT and MICT which contributed to effacing the superiority of HIIT and MICT to RRE. If this assumption stands up, it is easy to understand why an increment of TET effaced the superiority of HIIT to MICT in this meta-analysis.

However, each coin has its two sides. The result that an increment of TET effaced the superiority of HIIT to MICT also means that an increment of TET effaced the inferiority of MICT to HIIT. How interesting the result is! As is discussed above continuing, even if the adherence in MICT descended over time as the HIIT did, the situation of the increasing poor adherence over time was probably not balanced between HIIT and MICT, which was perhaps more serious in HIIT. That is to say the total energy expenditure between HIIT and MICT still became similar over time because of the imbalance of the increasing poor adherence between HIIT and MICT. This assumption is consistent with the previous meta-analysis by Gomes Neto et al. [36] showing that HIIT could improve Peak VO$_2$ compared with MICT, but its superiority disappeared if total energy expenditure was balanced between HIIT and MICT. Different from Gomes Neto et al. [36], this study emphasizes the importance of total energy expenditure from the view of TET and intensity and it might be helpful to ultimately understand how the "FITT" works in HIIT versus MICT through total energy expenditure. From the results and interpretation of this meta-analysis, the relationship between HIIT and MICT and the role of them in cardiac rehabilitation is just like an old story of Aesop's Fables about the turtle and the

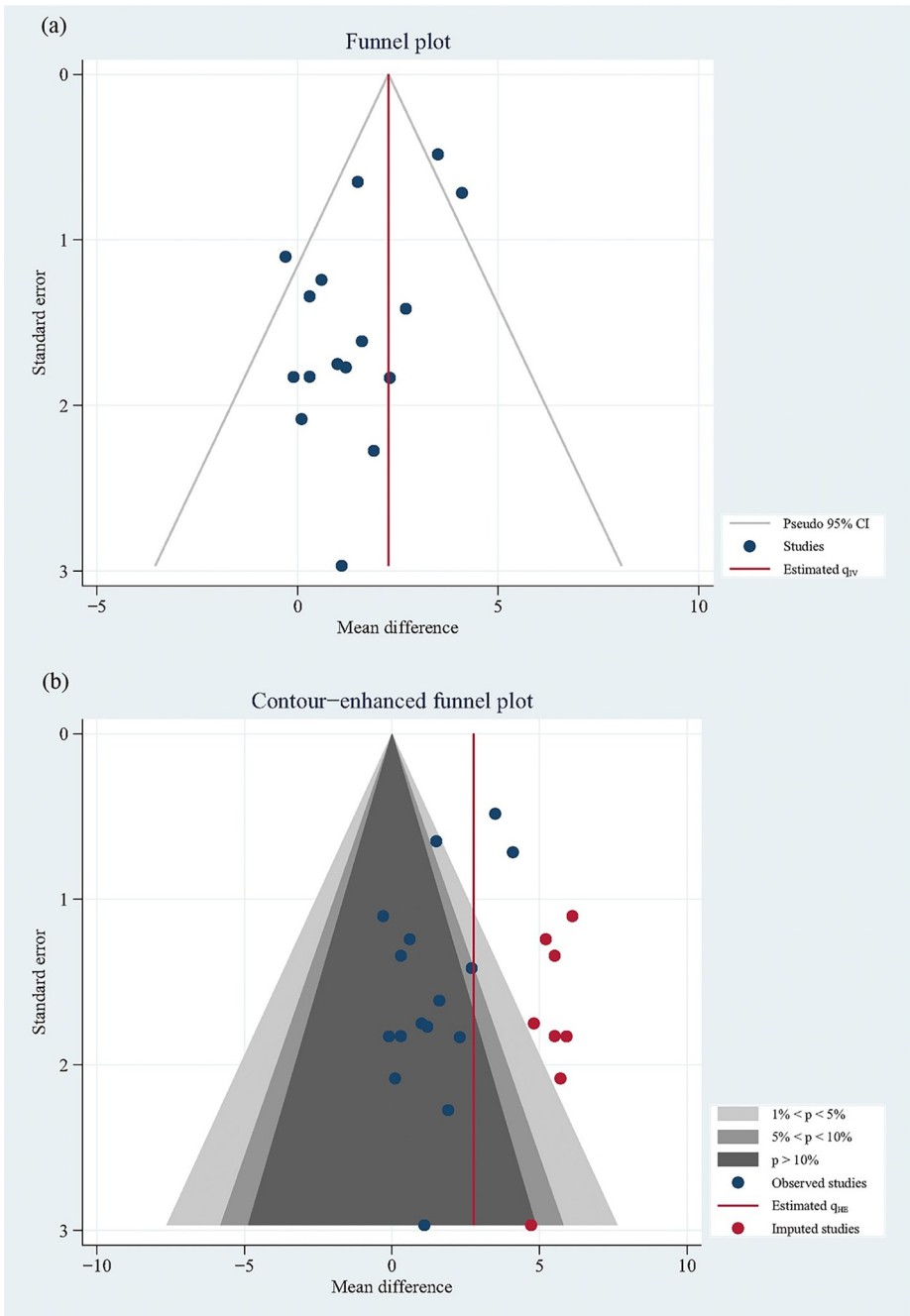

**Fig 7. Bias of publication.**

rabbit that had a race. HIIT is like the "rabbit" and MICT is the "turtle", and TET is the witness to this "race".

In this meta-analysis, we pooled HFpEF studies into analysis. And the results of subgroup analysis showed that it did not support HIIT was superior to MICT for HFpEF in improving Peak VO$_2$. An important caveat is that only three studies [10, 21, 25] of HFpEF were included, thus we could not perform the subgroup analysis of TET in HFpEF. So more randomized

clinical trials are needed to confirm if this "negative" conclusion was contributed by the poor adherence behind TET.

The strength of our study is that we have revealed a "paradox" of TET and made an attempt to explain it, which highlights the importance of TET and adherence to the target intensity for HF patients in a sports description in a cardiac rehabilitation program. For example, it is crucial for physician, cardiologist and physiotherapist to help patients recognize and address the practice barriers of the training protocol to achieve the prescribed exercise intensity. Nonetheless, it might be inadvisable to power the full focus on the exercise intensity with the neglect of TET. For instance, a patient with inevitable practice barriers of achieving the prescribed exercise intensity would still obtain the same benefit from exercise training by prolonging the TET. So patients would profit most from a realistic protocol balancing exercise time and intensity for a full energy expenditure. In a vivid word, if a patient in a cardiac rehabilitation program is the "rabbit", the key point to win the race is to keep his "speed" and if he is the "turtle", the opportunity to catch up with the "rabbit" depends on the "persistence". Certainly, this verdict needs more randomized and multicenter trials to confirm.

### 4.2. Limitations

Of the 16 studies, 14 were small size or/and single-center studies, which would limit the results of this meta-analysis. Thus, more studies conducted at multiple centers with with larger sample sizes and longer follow-up periods are needed to bolster the findings. Additionally, the contour-enhanced funnel plot showed the asymmetry of distribution of the studies in funnel plot was caused by some other potential reasons which might also limit the results of this meta-analysis, for example the different intervention characteristics of HIIT and MICT or the obvious difference between studies in the general characteristics including the ratio of men to women, mean age, and LVEF and Peak $VO_2$ at baseline et al. But complying with PRISMA guidelines in the process of this systematic review ensured the minimization of the biases involved in this systematic review.

### 5. Conclusion

Our meta-analysis showed a "paradox" that the superiority of HIIT to MICT in improving Peak $VO_2$ arose in a short to medium length of TET whereas it was effaced by an increment of TET perhaps due to the increasing poor adherence to target intensity over time. In addition, we found HIIT was not superior to MICT in VE/$VCO_2$ slope and QoL for HF patients. Our findings give an implication that new studies should focus on the key aspects of adherence and report more details on it. Promoting standardized measures such as self-report questionnaires, exercise logs, and wearable devices would help to objectively monitor adherence. Besides, our findings support the use of a realistic exercise training protocol balancing exercise time and intensity for a full energy expenditure in a cardiac rehabilitation program. An important caveat is that the negative effect of attenuation in adherence on HF patients was just indirectly proved by conducting a subgroup analysis of TET.

### Supporting information

**S1 Checklist. PRISMA 2020 main checklist.**
(DOCX)

**S1 File. Search strategy.**
(DOCX)

**S2 File. Descriptive statistical analysis of TET for HF and HFrEF with SPSS 21.0.**
(DOCX)

**S1 Fig. Sensitive analysis with STATA 16.0.** (a) for HF studies; (b) for HFrEF studies.
(TIF)

**S2 Fig. (a) Subgroup analysis of TET of HIIT for HFrEF in Peak VO$_2$; (b) sensitive analysis.**
(TIF)

**S3 Fig. (a) Subgroup analysis of TET of MICT for HFrEF in Peak VO$_2$; (b) sensitive analysis.**
(TIF)

**S4 Fig. VE/VCO$_2$ slope.**
(TIF)

**S5 Fig. Quality of life.**
(TIF)

## Author Contributions

**Conceptualization:** Xinchao Du.

**Data curation:** Shengyuan Gu, Dongwei Wang, Shifang Guo.

**Formal analysis:** Shengyuan Gu, Dongwei Wang, Shifang Guo.

**Funding acquisition:** Yaohua Yu.

**Investigation:** Dongwei Wang.

**Methodology:** Shengyuan Gu, Shifang Guo.

**Project administration:** Xinchao Du.

**Resources:** Dongwei Wang, Yaohua Yu.

**Software:** Shengyuan Gu, Yaohua Yu, Shifang Guo.

**Supervision:** Xinchao Du, Dongwei Wang, Yaohua Yu.

**Validation:** Xinchao Du, Yaohua Yu.

**Writing – original draft:** Shengyuan Gu, Xinchao Du.

**Writing – review & editing:** Xinchao Du.

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
