## [Decision Letter · Decision Letter 0]

5 Jul 2023

PONE-D-23-18099Effects of High Intensity Interval Training versus Moderate Intensity Continuous Training on Exercise Capacity and Quality of Life in Patients With Heart Failure: A Systematic Review and Meta-analysisPLOS ONE

Dear Dr. Du,

Thank you for submitting your manuscript to PLOS ONE. After careful consideration, we feel that it has merit but does not fully meet PLOS ONE’s publication criteria as it currently stands. Therefore, we invite you to submit a revised version of the manuscript that addresses the points raised during the review process.

We look forward to receiving your revised manuscript.

Kind regards,

Ahmed Mustafa Rashid

Academic Editor

PLOS ONE

Journal Requirements:

2. Please ensure that you include a title page within your main document. You should list all authors and all affiliations as per our author instructions and clearly indicate the corresponding author.

3. Please upload a new copy of Figures 4 and 5 as the detail is not clear. Please follow the link for more information: https://blogs.plos.org/plos/2019/06/looking-good-tips-for-creating-your-plos-figures-graphics/" https://blogs.plos.org/plos/2019/06/looking-good-tips-for-creating-your-plos-figures-graphics/

Reviewers' comments:

Reviewer's Responses to Questions

**Comments to the Author**

1. Is the manuscript technically sound, and do the data support the conclusions?

Reviewer #1: Yes

2. Has the statistical analysis been performed appropriately and rigorously? 

Reviewer #1: Yes

3. Have the authors made all data underlying the findings in their manuscript fully available?

Reviewer #1: Yes

4. Is the manuscript presented in an intelligible fashion and written in standard English?

Reviewer #1: Yes

5. Review Comments to the Author

Reviewer #1: In their study, Gu et al. conducted a comprehensive evaluation of the effects of high-intensity interval training (HIIT) compared to moderate-intensity continuous training (MICT) in individuals with heart failure. The importance of this study cannot be overstated, as optimizing exercise therapies in this population has significant implications for their overall well-being and management. However, to further enhance the study’s robustness the authors should consider incorporating the following points.

1. Page 1, Lines 11-17: Methods and results: it would be helpful if the authors could expand on the details of the electronic literature search, including the databases used and the number of studies included in the analysis. The authors should also consider reporting the effect sizes and confidence intervals for the subgroup analyses of TET to provide more precise information and improve the transparency of the study.

2. Page 10, Lines 35-39: The introduction provides a general overview of heart failure (HF) and its impact on quality of life. However, it would be helpful if the authors could include specific statistics to support the statements regarding the prevalence and burden of HF.

3. Page 11, Lines 42-45: The authors mention the benefits of exercise training (ET) for HF patients, but the introduction lacks a rationale for discussing the comparison between high-intensity interval training (HIIT) and moderate-intensity continuous training (MICT). It would be beneficial if the authors could provide a stronger justification for exploring the effectiveness of HIIT and MICT in HF patients.

4. Page 11, Lines 47-50: The authors have referred to the SMARTEX and OptimEx-Clin study, as both have made negative conclusions about the superiority of HIIT over MICT. It would be informative for the reader if the authors could briefly summarize the findings of these studies and their implications in a concise manner.

5. Page 11, Lines 53-56: The discussion of training and total exercise time (TET) is interesting, but it would be beneficial for the readers’ understanding if the authors could further clarify it. Consider providing a brief explanation of why training duration may not accurately represent the actual exercise time and how TET can provide a more comprehensive measure.

6. Pages 13-14, Lines 105-110: The authors should consider providing additional information regarding any disagreements or a process for resolving conflicts between reviewers while screening and selecting studies. This will improve the transparency and reliability of the study.

7. Page 14, Lines 112-114: The critical appraisal of included studies demonstrates an effort to assess the methodological quality of the studies. However, it would be valuable to report on the specific domains assessed and whether any measures were taken to mitigate the potential bias.

8. Page 13, Lines 91-92: The authors have mentioned the outcomes of interest Peak VO2, VE/VCO2 slope, and QoL. Please consider specifying how these outcomes were assessed and whether any standardized measurements were used.

9. Page 15, Lines 125-136: It is mentioned that Review Manager (Version 5.4), SPSS (Version 21.0), and STATA (Version 16.0) were used for data analysis. It would be informative if the authors could state the specific analyses conducted with each software. This would help enhance the transparency of the study.

10. Page 15, Lines 140-145: The authors have provided information on the number of articles identified, duplicates removed, and studies excluded at each stage of the screening process. However, the authors should consider providing specific reasons for excluding the studies as this would help improve the reliability of the study.

11. Page 16, Lines 168-172: The authors should expand on the interpretation of the sensitivity analyses conducted, specify why certain studies were identified as sources of potential heterogeneity, and explain why their exclusion affected the overall results. This could help increase the robustness of the results.

12. Page 17, Lines 180-182: It would be helpful if the authors could include the effect sizes along with the corresponding confidence intervals for all the reported outcomes. This could help enhance the interpretation of the results for the reader.

13. Page 17, Lines 193-195: Please provide additional information on the potential reasons for the asymmetry observed in the funnel plot, as this could help in better understanding the factors influencing the distribution of studies and improve the interpretation of publication bias in the research.

14. Page 20: The study explains how poor adherence over time can affect the total energy expenditure and the superiority and inferiority of HIIT and MICT. However, it would be beneficial if the authors could provide specific examples or evidence from the analyzed studies to strengthen these claims.

15. Page 22, Line 290: Please consider revising the sentence “limited by the scarcity of high-quality, large sample”. It would be helpful to clarify the specific characteristics that were lacking in the included studies. For instance, the authors could mention the need for studies with larger sample sizes, longer follow-up periods, and studies conducted at multiple centers to bolster the findings.

16. Page 22, Lines 292-296: Although the authors have mentioned the use of a counter-enhanced funnel plot, they haven’t provided an explanation for the observed asymmetry in the funnel plot. The authors should consider discussing the potential reasons for this asymmetry and its implications. Factors such as selective reporting or study design limitations could contribute to the observed funnel plot asymmetry.

17. Page 22 Lines 299-306: While the suggestion for future studies to report more details on adherence to HIIT or MICT is commendable, it would be valuable to provide specific recommendations to researchers. For instance, the authors could identify key aspects of adherence to focus on and promote standardized measures such as self-report questionnaires, exercise logs, and wearable devices to objectively monitor adherence.

18. Page 22 Lines 299-306: To provide a more comprehensible perspective for readers and facilitate proper interpretation of the findings, it would be beneficial to briefly address the limitations of the study. This will ensure that the readers interpret the findings in the appropriate context.

19. Page 11, Lines 51-52: The authors should consider rephrasing the phrase “In order to make it clear if the flowing time, behind which was the attenuation in adherence”. The sentence is hard to follow and might be ambiguous for the readers.

20. Page 11, Line 45: Please correct the spelling of the word trials.

6. PLOS authors have the option to publish the peer review history of their article (what does this mean?). If published, this will include your full peer review and any attached files.

Reviewer #1: No

---

## [Author Response · Author response to Decision Letter 0]

10 Jul 2023

Dear Editor,

 We would like to voice our great appreciation to you and the reviewer.

We thank you very much for giving us an opportunity to perfect our manuscript, of which the title is “Effects of High Intensity Interval Training versus Moderate Intensity Continuous Training on Exercise Capacity and Quality of Life in Patients With Heart Failure: A Systematic Review and Meta-analysis” with an ID “PONE-D-23-18099”, following the constructive suggestions of the reviewer. We also appreciate the volunteer reviewer for his positive comments to our work and for his helpful, constructive and meticulous suggestions. 

Thank you!

Looking forward to hearing from you.

Yours sincerely,

Xinchao Du on behalf of the authors.

---

## [Decision Letter · Decision Letter 1]

7 Aug 2023

Effects of High Intensity Interval Training versus Moderate Intensity Continuous Training on Exercise Capacity and Quality of Life in Patients With Heart Failure: A Systematic Review and Meta-analysis

PONE-D-23-18099R1

Dear Dr. Du,

We’re pleased to inform you that your manuscript has been judged scientifically suitable for publication and will be formally accepted for publication once it meets all outstanding technical requirements.

Kind regards,

Ahmed Mustafa Rashid

Academic Editor

PLOS ONE

Additional Editor Comments (optional):

Reviewers' comments:

Reviewer's Responses to Questions

**Comments to the Author**

1. If the authors have adequately addressed your comments raised in a previous round of review and you feel that this manuscript is now acceptable for publication, you may indicate that here to bypass the “Comments to the Author” section, enter your conflict of interest statement in the “Confidential to Editor” section, and submit your "Accept" recommendation.

Reviewer #1: All comments have been addressed

2. Is the manuscript technically sound, and do the data support the conclusions?

Reviewer #1: Yes

3. Has the statistical analysis been performed appropriately and rigorously? 

Reviewer #1: Yes

4. Have the authors made all data underlying the findings in their manuscript fully available?

Reviewer #1: Yes

5. Is the manuscript presented in an intelligible fashion and written in standard English?

Reviewer #1: Yes

6. Review Comments to the Author

Reviewer #1: (No Response)

7. PLOS authors have the option to publish the peer review history of their article (what does this mean?). If published, this will include your full peer review and any attached files.

Reviewer #1: No

---

## [Editor Report · Acceptance letter]

9 Aug 2023

PONE-D-23-18099R1 

Effects of High Intensity Interval Training versus Moderate Intensity Continuous Training on Exercise Capacity and Quality of Life in Patients With Heart Failure: A Systematic Review and Meta-analysis 

Dear Dr. Du:

I'm pleased to inform you that your manuscript has been deemed suitable for publication in PLOS ONE. Congratulations! Your manuscript is now with our production department. 

Kind regards, 

on behalf of

Dr. Ahmed Mustafa Rashid 

Academic Editor

PLOS ONE